# Nanoparticle-Based Approaches towards the Treatment of Atherosclerosis

**DOI:** 10.3390/pharmaceutics12111056

**Published:** 2020-11-05

**Authors:** Artur Y. Prilepskii, Nikita S. Serov, Daniil V. Kladko, Vladimir V. Vinogradov

**Affiliations:** International Institute “Solution Chemistry of Advanced Materials and Technologies”, ITMO University, 191002 Saint Petersburg, Russia; prilepskii@scamt-itmo.ru (A.Y.P.); serov@scamt-itmo.ru (N.S.S.); kladko@scamt-itmo.ru (D.V.K.)

**Keywords:** atherosclerosis, nanoparticles, target delivery

## Abstract

Atherosclerosis, being an inflammation-associated disease, represents a considerable healthcare problem. Its origin remains poorly understood, and at the same time, it is associated with extensive morbidity and mortality worldwide due to myocardial infarctions and strokes. Unfortunately, drugs are unable to effectively prevent plaque formation. Systemic administration of pharmaceuticals for the inhibition of plaque destabilization bears the risk of adverse effects. At present, nanoscience and, in particular, nanomedicine has made significant progress in both imaging and treatment of atherosclerosis. In this review, we focus on recent advances in this area, discussing subjects such as nanocarriers-based drug targeting principles, approaches towards the treatment of atherosclerosis, utilization of theranostic agents, and future prospects of nanoformulated therapeutics against atherosclerosis and inflammatory diseases. The focus is placed on articles published since 2015 with additional attention to research completed in 2019–2020.

## 1. Introduction

Atherosclerosis is a chronic inflammatory disease of the arterial wall caused by dyslipidemia and regulated by the innate and adaptive immune response. For many years, it was believed that high blood pressure, as well as high blood cholesterol levels, were the main factors influencing the pathogenesis of atherosclerosis [1]. Currently, the research data convincingly demonstrates that the inflammatory process is central during all stages of the development of this disease [2]. The artery wall consists of three layers: inner (intima), middle (media), and outer (adventitia). The intima is represented by one layer of endothelial cells, a thin basement membrane, and a subendothelial layer of collagen. The media is formed by vascular smooth muscle cells (VSMCs), and a network of elastic and collagen fibers. Adventitia is the outer layer of connective tissue. Additionally, atherosclerosis primarily develops in regions exposed to turbulent blood flow, while in regions with laminar flow the occurrence of plaques is very low [3]. Low shear stresses are also a major cause of plaque formation [4].

A characteristic feature of atherosclerosis is the formation of an atherosclerotic plaque in the intima, proliferation of VSMCs, as well as the accumulation of activated immune cells, and the proliferation of adventitia in the area of plaque formation (Figure 1) [5]. Inflammation and biochemical modifications causing endothelial and VSMCs proliferation, produce extracellular matrix molecules and form a fibrous cap over the developing plaque [6]. Many immune system cells are present in the arterial wall but their number increases significantly as atherosclerosis progresses. Under normal conditions, immune cells migrate into the wall of a healthy vessel and return to the blood flow [7].

In the early stages of atherosclerosis, low-density lipoproteins (LDL) accumulate in the aortic wall with subsequent oxidation (oxLDL). The accompanying increase in blood pressure leads to the activation of the endothelium, expression of adhesion molecules growth, and to the recruitment of monocytes into the intima [8]. Monocytes further differentiate into macrophages and absorb oxLDL, thereby forming lipid-filled foam cells. Activated macrophages produce cytokines thus promoting the influx of pro-inflammatory cells into the vessel wall. The ability of macrophages to leave the inflammation site is reduced, which leads to their accumulation in the atherosclerotic plaque and adjacent adventitia. Meanwhile, the proliferation of macrophages is significantly increased for both resident and monocyte-derived types [9]. A distinctive feature of the late stages of atherosclerosis is the progressive accumulation of foam cells in the atherosclerotic plaque. Foam cells die mainly due to necrosis and form the “so-called” necrotic core of the plaque. The necrotic core destabilizes the structure of the plaque and promotes its rupture and thrombus formation [10]. According to several autopsy studies, approximately 60% of acute myocardial infarctions occur as a result of the rupture or fissuring of thin-capped fibroatheroma [11].

The most common animal models for studying the development of atherosclerosis are mice with a genetic knockout of the apolipoprotein E gene (ApoE^−/−^) or the gene encoding the LDL receptor (Ldlr^−/−^). ApoE^−/−^ mice are susceptible to spontaneous development of atherosclerosis, which can be aggravated by keeping them on a high-fat diet. At the same time, in Ldlr^−/−^ mice, atherosclerosis is induced only by a high-fat diet. However, human advanced plaques are much more complex structures, which comprise inorganic components. Some novel in vivo models are under development to better represent human behavior [12].

The main atherosclerosis treatment strategy, for now, is still the surgical removal of plaque or stent implantation. Since the discovery of the leading role of cholesterol in atherosclerosis development, lipid control was the subject of research for many years. The subsequent development of statin drugs started a new era in atherosclerosis treatment. However, despite the large-scale use of statins, their side effects-to-effectiveness balance is far from perfection. Several pharmacological strategies with varying degrees of potency were developed in recent years [13]. However, only limited efficacy of statins has been observed in clinical conditions where the drugs are administered systemically, which is likely attributed to the rapid drug clearance and unsatisfactory accumulation at the arterial injury site [14]. Nanoformulated drugs are a good choice for atherosclerosis treatment due to a reason that is usually considered as a negative effect: most nanoparticles (NPs) are consumed by macrophages. As NPs undergo phagocytosis by macrophages, sophisticated concepts are being developed to protect NPs from phagocytosis. In the case of atherosclerosis, macrophages are naturally present in abundance inside the plaque. Thus, phagocytosis can be considered as an innate way to deliver nanoformulated drugs to a plaque inside macrophages. This and other in vivo strategies of drugs target delivery will be addressed in this review.

## 2. Nanocarriers-Based Drug Targeting Principles

Atherosclerosis shares many similarities with the tumor metastasis process [4]. Angiogenesis and cell migration during atherosclerosis formation and tumor growth have almost the same biochemical markers [15]. Thus, many anticancer drugs and approaches have found their application as anti-inflammatory agents. The affected endothelium presents many specific receptors on its surface, that become suitable targets for highly specific targeting [16,17]. Even though macrophages are present in many tissues and participate in various processes, they can be found in abundance in atherosclerotic plaques, thus they are becoming a reasonable target for atherosclerotic lesions [18].

### 2.1. ανβ3 Integrin

Integrins are heterodimeric membrane glycoproteins composed of α and β subunits. Integrins can bind to different components of the surrounding extracellular matrix. Among many subfamilies, there is ανβ3 integrin that plays a crucial role in angiogenesis and metastatic dissemination. This integrin is expressed on endothelial cells and upregulated by pro-angiogenic growth factors [19]. For instance, the cRGDfK peptide can specifically bind to ανβ3 which was utilized in [20].

### 2.2. Stabilin-2

The stabilin-2 (STAB2, HARE) receptor is a transmembrane receptor that participates in processes of angiogenesis, cell adhesion, and some others. This receptor is known for its ability to bind and clear multiple ligands, among which are heparin and hyaluronic acid (HA) [21]. It was found that this receptor is profusely expressed on the surface of macrophages and endothelium cells in atherosclerotic plaques [22]. HA is highly biocompatible and biodegradable and was successfully utilized as a capping and target agent in cancer treatment. Self-assembled HA nanoparticles were used to visualize plaques both through binding to macrophages, and diseased endothelium [23]. Furthermore, the synthetic peptide, CRTLTVRKC (S2P), which is specifically bound to STAB2 has been synthesized [22]. 

### 2.3. CD44

CD44 is a well-studied glycoprotein that is involved in cell–cell interactions, cell adhesion of lymphocytes to the endothelium, and activation of inflammatory and vascular cells [24]. CD44 inhibition was found to reduce the recruitment of T lymphocytes to lesions and decrease the activation of VSMCs [25]. However, CD44 can possess pro-atherogenic behavior by inducing the production of soluble inflammatory mediators [26]. HA, mentioned above, is a principal ligand of CD44 [27]. NADPH oxidase-mediated reactive oxygen species (ROS) generation enhanced thrombin-induced HA synthesis which, in turn, leads to an increase in low molecular weight HA and VSMCs proliferation [28].

### 2.4. IL-1 Receptor Antagonist (IL-1ra)

Cytokine interleukin IL-1 is one of the main pro-inflammatory factors expressed in the endothelium of atherosclerotic plaques [29]. IL-1 is naturally inhibited by IL-1ra and was proven to be effective for the treatment of different inflammatory diseases [30,31]. However, the role of IL-1 in atherosclerosis is rather complicated. Controversial data on IL-1 role in plaque stability states that it can play both protective and destabilization roles. Subcutaneous injection of recombinant IL-1ra leads to a decrease in the lesion area, reduces the number of foam cells in the plaques, and overall plaque stability [32]. In contrast, Alexander et al. showed that the inactivation of the IL-1 signaling pathway resulted in plaque instability, reduced matrix metallopeptidase 3 (MMP 3) levels, and lowered migration of VSMCs [33]. Thus IL-1 appears to be a hindered target in atherosclerotic treatment. 

### 2.5. Vascular Cell Adhesion Molecule-1 (VCAM-1)

VCAM-1 and ICAM-1 (intercellular cell adhesion molecule-1) are a part of cytokine-induced immunoglobulins (IGs) that bind leukocyte integrins a4 and b2 [34]. Both VCAM-1 and ICAM-1 are overexpressed in vessel regions undergoing shear stress [35]. ICAM-1 was found to be in close relation with plasma cholesterol levels [36]. VCAM-1 expression is increased in patients with already developed lesions while circulating ICAM-1 is a predictor of future events [37].

### 2.6. LDL Targeting

Since low-density lipoproteins accumulate inside atherosclerotic plaques, it seems promising to use them as naturally occurring targeting agents. In 2017, Sobot et al. proposed to use cholesterol precursor squalene, conjugated with drugs, to target LDL-accumulating cancer cells [38,39]. Further investigation brings forward the opportunity to use squalene as a guiding molecule for atherosclerosis targeting. For now, only imaging modality with fluorescent markers has been established [40]. However, the potential of this approach could be prominent.

## 3. Methods for the Synthesis of Particles

There is a huge number of synthesis methods of nanoparticles for biomedical applications. Since several classes of nanoparticles are used to design nanoformulated therapeutics, in this section we briefly overview the most common synthesis procedures of organic (liposomal, polymer-based, and solid lipid) and inorganic (metal and metal oxide) nanoparticles. In addition to the synthetic approaches listed and described below, additional particle modifications could be introduced including: core-shell structures formation [41], electrostatic [42], and covalent [43] surface modification, self-assembled suprastructures formation [44,45,46,47,48]. A large number of micro- and nano-sized systems for biomedical applications including atherosclerosis treatment could be produced by following these procedures.

### 3.1. Organic Particles

Organic nanoparticles typically consist of small organic or/and polymeric molecules; large structural and chemical variability of which allows one to obtain custom drug delivery systems (DDSs) for particular drugs and biomedical applications. Liposomes, for example, are proven to be effective in hydrophilic and hydrophobic drugs delivery, while still being easy to design and synthesize [49]. Polymeric nanoparticles have shown tunable bloodstream circulation times, toxicity, and drug release kinetics [50]. Solid lipid particles, in turn, are much more suitable for hydrophobic drugs and offer significant control over particle size and polydispersity as well as drug release kinetics. Thus, organic particles represent a suitable platform for the treatment of atherosclerosis and inflammation-related diseases.

#### 3.1.1. Liposomes

Liposomes, despite their long history, are still one of the most commonly used drug delivery vehicles. Many methods have been developed in recent years to produce more predictable and smaller sized liposomes [51] which are listed and discussed below.

The filming-rehydration method is used quite extensively to produce liposomes for medical applications [20,46,52,53]. Usually, a mixture of phospholipids and modified polymers is used to form a stable lipid bilayer. Two miscible solvents (for instance, chloroform and methanol in the proportion of 2:1 vol.) are chosen next. Phospholipids and polymers are then dissolved in this biphasic solvent system followed by vacuum evaporation of the organic solvent and subsequent thin film formation. The resulting film is rehydrated by incubation with a solution containing molecules of interest. Obtained liposomes can be modified by ultrasonication with additional phospholipidic components dissolved in saline.

Emulsification is another frequently used method for liposomes production [54,55]. The process of emulsification, in contrary to the filming-rehydration approach, allows one to circumvent the step of film formation. Homogenization of a molecule-containing solvent mixture is followed by its extraction to the organic phase, the addition of dried phospholipids and distilled water, and subsequent emulsification of the mixture. Next, the organic phase is excluded by vacuum evaporation and obtained solid lipid particles are extruded to form nano-sized ones with a smaller size distribution. This method allows to precisely control the size of formed solid lipid particles and to reach high loading efficacies, although it is not suitable for hydrophilic molecules. 

#### 3.1.2. Polymer-Based Particles

Flash nanoprecipitation is simple, scalable, and the most common process utilized for the production of polymer-based particles. Nanoprecipitation is based on rapid micromixing which creates high supersaturation conditions leading to the precipitation and encapsulation of hydrophobic drugs in a polymer-based delivery vehicle [56]. The flash nanoprecipitation was extensively used for nanocarrier for atherosclerosis therapy design [44,48,57]. Wang et al. designed the carrier via covalent conjugation of a superoxide dismutase mimetic agent Tempol and a hydrogen peroxide-eliminating compound of phenylboronic acid pinacol ester onto a cyclic polysaccharide β-cyclodextrin (Table 1) [57]. Lewis et al. created the sugar-based amphiphilic macromolecules to competitively block oxidized lipid uptake via scavenger receptors on macrophages [44]. Dou with co-authors used a facile synthesis of biocompatible materials from α- or β-cyclodextrin and their polymers to load and deliver anti-miR33 into target cells [48].

Protein-based nanoparticles also can be considered as polymeric particles. The main advantage of these particles is their excellent biocompatibility [58]. Protein nanoparticles can be prepared from collagen, keratin, silk, soy protein, corn zein, elastin, etc. [59]. Especially attractive, is the synthesis of nanoparticles from albumins, namely bovine albumin [60,61] or human albumin [62,63].

#### 3.1.3. Solid Lipid Particles

Solid lipid particles possess a solid lipid core matrix consisting of glycerol derivatives which can in turn encapsulate hydrophobic drugs. This core is usually stabilized with cationic, anionic, neutral, or zwitterionic surfactants depending on the particular goal. In case of biomedical applications, solid lipid particle synthesis procedures usually do not utilize organic solvents, which is always desirable for nanoformulated drugs. Several techniques have allowed the obtaining of micro- and nano-sized solid lipid particles containing volatile and non-volatile hydrophobic drugs.

High shear homogenization and ultrasound treatment are dispersing techniques that were initially used for the production of solid lipid particles [64]. Notwithstanding the approach simplicity, it is often difficult to remove micro-sized particles and achieve high polydispersity, thus the yield of nano-sized solid lipid particles is usually low.

High-pressure homogenization [65] represents another approach that is usually utilized for the production of nanoformulated drugs for parenteral nutrition. One of the main technique’s advantages is its scalability, thus large scale production of solid lipid particles can be achieved. Briefly, homogenizers push a liquid with high pressures of up to 2000 bar through a narrow micron-sized gap. Very high shear stress, as well as cavitation forces, disrupt particles down to the submicron range.

The hot homogenization approach deals with emulsions of lipids kept above their melting temperatures. The increase in temperature lowers the viscosity of lipids, thus particles of smaller sizes can be obtained using this technique; although it is not suitable for volatile drugs that degrade upon heating.

Solvent emulsification/evaporation technique [66] is often used to obtain solid lipid nanoparticles by evaporation of non-polar water-immiscible solvents from o/w lipid- and drug-containing emulsions. It was shown that solid lipid nanoparticles smaller than 50 nm can be obtained through this procedure [67].

### 3.2. Inorganic Nanoparticles

#### 3.2.1. Metal Particles

Metal nanoparticles are attractive carriers for gene delivery due to their bioinert properties, low cytotoxicity. A large number of protocols existing [68] for the tuning of particle size, shape, and for functionalization of particles by different ligands [69,70].

Chemical reduction represents the most common method of metal-derived NP preparation due to its simplicity and convenience [71,72]. Certain chemicals of artificial (e.g., hydrogen, sodium borohydride, hydrazine hydrate, glucose, sodium citrate, ethylene glycol, ethanol) or natural (lignin) origin can reduce metal ionic salts [73]. The formation of colloidal NP solutions is a two-stage process which includes nucleation and crystal growth with subsequent formation of NPs of various sizes, shapes as well as surface charges. Usually, the stabilizer or capping agent, such as sodium lauryl sulfate, sodium oleate, polyvinyl alcohol, or polyvinylpyrrolidone, is required to stabilize the sol [74].

Electrochemical deposition represents a voltage-dependent technique when NPs produced from metal salts solution are deposited on the cathode [75]. This method is widely used in cyclic voltammetry, double pulse, and potential step deposition.

The photochemical method allows metal atom formation without precipitation. Metal atoms can be generated via direct or indirect photoreduction which use photochemically generated intermediates, e.g., radicals, to reduce metal ions [76]. The indirect route is advantageous as it is less time-consuming and allows for the fine-tuning of NP parameters, including excitation wavelength [77].

The seed-mediated growth method is a typical example of a heterogeneous nucleation process for the synthesis of gold nanoparticles. A typical growth process involves two steps: the synthesis of seed nanoparticles and their subsequent growth in solutions containing metal precursors, reducing reagents, and shape-directing reagents [78]. Following these techniques, Sun et al. [79] synthesized—via seed-mediated method—Au nanoparticles for further VCAM1 and anti-miR-712 immobilization and delivery to target endothelial cells.

Green synthesis has gained increased popularity in recent years, particularly due to the unsafe chemical or physical approaches for nanoformulations and biomedical applications [80]. In this route, NPs are produced via two principal directions: bioreduction or biosorption. By the bioreduction method, the metal ions are chemically reduced by bacterial enzymes, and the synthesized NPs can be easily removed from the reaction. The bioinspired strategies imply the production of NPs by living organisms such as plants [81,82], bacteria [83], and fungi [84], as well as the use of biological extracts and enzymes as a primary source of precursors [85,86].

#### 3.2.2. Metal-Oxide Particles

Despite their inorganic nature, the toxicity of several metal oxides was found to be low [87,88]. Biocompatibility of magnetite-based and aluminum oxide-based nanoformulation allows their use in different biomedical approaches, including the delivery of enzymes [89,90], bioimaging, and therapy. Magnetite is one of the most attractive magnetic inorganic metal-oxides due to its biocompatibility, relatively high magnetic properties, and a wide range of synthetic procedures [91]. The main synthetic procedures include the co-precipitation [92,93], hydro- and solvothermal syntheses [94,95], reduction in template iron oxide with desired shape and size (for example, α-FeOOH, β-FeOOH, α-Fe_2_O_3_) in wet (polyol reduction [96], hydrazine [97]) and dry reduction [98] (mixed atmosphere of Ar and H_2_), and thermal decomposition of iron precursors [99].

Co-precipitation [100] occurs in aqueous media where iron ions form a hexahydrate complex. Depending on the oxidation state and pH, the iron complex can be involved in hydroxylation reactions to form the iron oxide nanoparticles.

The main concept of hydrothermal and solvothermal synthesis is the crystallization of compounds from high-temperature solutions at high vapor pressures. The reaction course strongly depends on the solubility of these compounds. A temperature gradient is maintained during this process, where at the hotter end the solute is dissolved and at the cooler one—deposited on a seed crystal. The synthesis is usually implemented in an autoclave, followed by its heating in an oven. 

Solvothermal synthesis allows one to obtain crystalline phases which are not stable at the melting point. The main disadvantage of this method is the inability of real-time observations of crystal growth. Polyol conditions act as a solvent, surfactant, and as a reducing agent. It is known that polyols reduce metal salts to metal nuclei, which then crystallize to form metal particles [101]. In addition to excellent colloidal stabilization, the high boiling point of the polyols is an advantage as this allows particles to be synthesized at 150–320 °C, without the need for high pressure and autoclaves.

## 4. Approaches towards the Treatment of Atherosclerosis

Nowadays, a healthy diet and lifestyle remain the main ways to prevent plaques formation; however, they do not guarantee absolute effectiveness due to many reasons. Medicinal atherosclerosis treatment methods can be divided into two groups: preventing plaque growth in early stages or stabilizing/reducing existing plaques. Prevention methods can include lowering the overall cholesterol level and enhancing efferocytosis in the early stages of plaque formation. Advanced plaques also can benefit from enhanced efferocytosis due to the reduction in necrotic core size and lowered cytokine levels. Despite the fact, that some types of NPs are known for the pro-atherogenic effect [102,103,104], many nanoformulations are currently being tested in vivo without any negative undesired effects on vessel structure. Nevertheless, such adverse consequences always should be considered when choosing a proper drug carrier. At the same time, some nanoparticles have been reported to have antioxidant properties on human umbilical vein endothelial cells (HUVECs), inducing autophagy instead of apoptosis [105]. The main strategies to prevent atherosclerotic manifestations that appeared in recent years will be discussed further. A summary of the discussed results is shown in Table 1 and Table 2, representing data obtained in vitro and in vivo.

**Table 1 pharmaceutics-12-01056-t001:** Nanoformulation with in vitro proved efficacy.

Model	NPs Composition	NPs Size	Drug Load	Targeted Modality	Results	Ref.
Human monocyte-derived macrophages	Amphiphilic polysaccharide, mucic acid	100–400 nm	-	Self-bind to scavenger receptors MSR1 and CD36	Inhibition of oxLDL uptake by macrophages.	[57]
Immortalized murine aortic endothelial cells	Au nanospheres	5, 10, 20, 50 nm	Anti-miR-712	VCAM-1 targeting peptide	Shown internalization of NPs by cells. The best size for accumulation	[79]
RAW264.7 (transformed into inflammatory and foam cells) and HUVECs cells	Oxidation-sensitive chitosan oligosaccharide nanoparticles coated/not coated in macrophages membrane	∼204 to ∼227 nm	Atorvastatin	Phagocytosis	No signs of cytotoxicity. Better viability of diseased macrophages after treatment with NPs. Reduced NO production and apoptosis. Coating NPs in macrophages membranes effectively reduces uptake by macrophages.	[106]
RAW264.7 cells	Cyclodextrin NPs coated with phospholipids	~100 nm	Simvastatin	Phagocytosis	Dissolution of cholesterol crystals inside cells. Reduced cholesterol levels in the media. Reduced secretion of MCP-1 and TNF-α. Inhibition of cell proliferation.	[45]
HUVECs and L929 cells	Apoptotic body biomimetic liposomes	90 to 140 nm	Pioglitazone	ανβ3 integrin targeting cRGDfK peptide	Reduced expression of IL-1β, IL-6, and TNF-α, shifting of macrophages phenotype from M1 to M2.	[20]

NPs: nanoparticles; MSR1: Macrophage scavenger receptor 1; CD36: cluster of differentiation 36; oxLDL: oxidized low-density lipoprotein; VCAM-1: vascular cell adhesion molecule 1; HUVECs: human umbilical vein endothelial cells; MCP-1: monocyte Chemoattractant Protein 1; TNF-α: tumor necrosis factor α; IL-1β: interleukin-1β; IL-6: interleukin-6.

**Table 2 pharmaceutics-12-01056-t002:** Nanoformulation with in vivo proved efficacy.

Model	NPs Composition	NPs Size	Drug Load	Targeted Modality	Results	Ref.
Ldlr^−/−^ mice	PLGA-PEG	<100 nm	Ac2-26 peptide	Collagen IV	Increase in collagen layer, a decrease in the necrotic core size, reduced oxidative stress.	[107]
ApoE^−/−^ mice	Amphiphilic polysaccharide, mucic acid	100–400 nm	-	Self-bind to scavenger receptors MSR1 and CD36	Reduced inflammation through lowered lipid content, neointimal hyperplasia, and inflammatory signaling. Reduced necrotic core size.	[57]
ApoE^−/−^ mice	Cationic lipoparticles	144 ± 55 nm	Anti-miR-712	VCAM1 targeting peptide	Significantly reduced lesion development.	[54]
C57BL/6 mice	Au nanospheres	5, 10, 20, 50 nm	Anti-miR-712	VCAM-1 targeting peptide	Au nanoparticles with a size of 5 nm have the best accumulation rate in the left carotid artery.	[79]
Ldlr^−/−^ mice	PLGA-*b*-PEG	156.6 ± 10.3 nm	Synthetic LXR agonist GW3965	Phosphatidylserine (optional)	Suppressed TNFα and MCP-1 levels. Reduced total cholesterol level in blood. Reduction in macrophage content in plaques. 50% reduced inflammation and lesion area.	[46]
ApoE^−/−^ mice	NPs from cyclic polysaccharide β-cyclodextrin	128 ± 1 nm	Tempol and phenylboronic acid pinacol ester	Phagocytosis by macrophages	Decrease in macrophage content and amount of cholesterol crystals. Decrease in the necrotic core volume and lowered ROS accumulation. Suppressed MMP 9 expression.	[44]
ApoE^−/−^ mice	High-density lipoprotein nanoparticle	<220 nm	Simvastatin	Phagocytosis by macrophages	43% reduced plaque size. Reduced macrophages proliferation and numbers by 65%. Suppressed plaque inflammation by silencing pro-inflammatory genes.	[108]
ApoE^−/−^ mice	Single-walled carbon nanotubes	5–6 nm in diameter, >60 nm in length	Src homology 2 domain-containing phosphatase-1 inhibitor	Phagocytosis by Ly-6Chi monocytes	Promoted efferocytosis resulting in the reduced necrotic core, lesion area, and debris amount.	[109]
ApoE^−/−^ mice	Oxidation-sensitive chitosan oligosaccharide nanoparticles coated in macrophages membrane	∼204 to ∼227 nm	Atorvastatin	Phagocytosis by macrophages	Reduced plaque area compared to the free drug (8% vs. 15%). Reduced the number of monocytes and MMP 9 levels. Thicker fibrous cap and increased proliferation of VSMCs, leading to overall plaque stability. General reduction in inflammation. Macrophages membranes were found to remove inflammatory cytokines or chemokines. Inhibited neovessel endothelial proliferation.	[106]
ApoE^−/−^ mice	Cyclodextrin NPs coated with phospholipids	~100 nm	Simvastatin	Phagocytosis by macrophages	Prolonged circulation time in blood and accumulation within plaque compared to free statin. Reduced proliferation of macrophages and plaque cholesterol levels. Plaque growth inhibition in the early stages of formation. Regression of existing plaques but no impact on blood cholesterol levels effect was observed.	[45]
Ldlr^−/−^ mice	PLGA core with a lipid-PEG shell	116.2 ± 2.5 nm	siRNA against Camk2g gene	S2P peptide, targeting macrophage stabilin-2 receptor	Significant reduction (2–3 times) in CaMKIIγ level proving the silencing of the corresponding gene by siRNA. Efferocytosis promotion. About 20% reduction in necrotic core volume. Twice increased fibrous cap thickness. Overall reduced lesion area.	[47]
ApoE^−/−^ mice	α-Cyclodextrin based pH-sensitive NPs	147.5 ± 2.1 nm	miR-33	Cyclic pentapeptide (cRGDfK) targeting ανβ3 integrin	Promotion of cholesterol efflux from macrophages. Reduced necrotic core. Increased VSMCs content.	[48]
Fat-feed New Zealand white rabbits	Janus particles with a silica core and covered in platelet membrane shell	300–400 nm	Paclitaxel	Direct delivery, anti-VCAM-1 antibody	Elimination of inflammatory macrophages, long-term anti-proliferation effect	[95]
ApoE^−/−^ mice	Apoptotic body biomimetic liposomes	90–140 nm	Pioglitazone	ανβ3 integrin targeting cRGDfK peptide	Significantly decreased expression of IL-1β and TNF-α due to reduced numbers of M1 macrophages in plaque. Increased collagen amount in fibrous cap, slightly reduced plaque area.	[20]

NPs: nanoparticles; PLGA: poly(lactic-co-glycolic acid); PEG: poly(ethylene glycol); MSR1: Macrophage scavenger receptor 1; CD36: cluster of differentiation 36; VCAM-1: vascular cell adhesion molecule 1; LXR: liver X receptor; ROS: reactive oxygen species; MMP 9: matrix metallopeptidase 9; VSMCs: vascular smooth muscle cells; CaMKIIγ: Ca2+/calmodulin-dependent protein kinase II-γ; MCP-1: monocyte Chemoattractant Protein 1; TNF-α: tumor necrosis factor α; IL-1β: interleukin-1β.

### 4.1. Angiogenesis Prevention

Angiogenesis is a well-known companion of atherosclerosis progression. Accordingly, targeting specified integrins can accurately mark the plaques or help in delivering the medicines. Antiangiogenic drugs are commonly combined with integrin-targeting moieties. Fumagillin is well known for its antimicrobial and amebicide activity, as well as for its angiogenesis inhibition [110]. It was primarily used in antitumor treatment but some research utilizes its ability to decrease vasa vasorum proliferation. Superparamagnetic nanoparticles were conjugated with complex ligands to target ανβ3 integrin and were loaded with fumagillin during synthesis [111]. Many anticancer drugs possess the ability to drastically decrease the proliferation of cells. It was proposed that such drugs as paclitaxel [112], methotrexate [113], or docetaxel (DTX) [114] can prevent numerous atherosclerotic manifestations. The research group of professor Raul Maranhão achieved significant progress in the development of lipid-based nanoparticles (LDE) aimed at the regression of atherosclerotic plaques [55,114,115,116]. Nanoformulation of DTX as solid lipid particles with an average diameter of 45–60 nm considerably lowers its cytotoxicity and side effects. Concurrently, it leads to a beneficial decrease in the main pro-atherogenic parameters, for instance, atheroma area reduction by 80%. Direct delivery of paclitaxel-loaded nanocapsules on the surface of the balloon catheter was proposed in [95]. The composition of particles was rather complex, possessing Janus-like behavior (Table 2). The core of the particles was formed from porous amine-modified silica, covered in the platelet membrane, and doped with anti-VCAM-1 antibody. Additionally, one side of the particles was covered with Pt to provide IR light-induced stimuli movement ensuring penetration of particles inside the plaque.

### 4.2. PDGF Receptor Inhibitor

Platelet-derived growth factor (PDGF) signaling pathways are involved in many malignant transformations, as well as in pathological mesenchymal responses, including atherosclerosis, restenosis, and fibrotic diseases [117]. PDGF is expressed in almost all cell types of the atherosclerotic region. The exact pathways for increased expression of PDGF (especially of A and B type) are poorly understood but mainly caused by hypercholesterolemia [118], reduced endothelial shear stress [118], or increased blood pressure [119]. PDGF is also expressed on the surface of VSMCs and causes their accumulation in the intima [120,121]. Based on these facts, inhibitors of PDGF can be promising to reduce lesions and progression of plaques growth [122]. In a very recent article [123], the widely used anticancer drug, imatinib, was encapsulated inside PLGA nanoparticles (with an average diameter of 183 nm) following conjugation with an S2P peptide. While the loading rate of nanoparticles was only 5%, the release rate was almost 100% over 5 days. Imatinib along with ponatinib and nilotinib is currently approved only for the treatment of leukemia and some stromal tumors by inhibiting the Bcr-Abl tyrosine kinase [124]. However, only imatinib has a beneficial effect on the cardiovascular system, decreasing plasma cholesterol levels, reducing the atherosclerotic lesion, and improving plaque morphology. Ponatinib and nilotinib, on the other hand, increase the gene expression of coagulation factors VII and VIIa, revealing pronounce pro-thrombotic effects [125]. Imatinib was found to prevent macrophage transformation into foam cells by decreasing cholesterol levels. Additionally, it causes the activity lowering of metalloproteinases MMP 2 and MMP 9 which are known to cause rapture of fibrin cap on atherosclerotic plaques [126]. Considering the various adverse effects of imatinib, targeted delivery of that drug would be beneficial.

### 4.3. Protection of VSMCs

VSMCs constitute the main part of the tunica media and therefore play a significant role in the overall integrity of the vessel wall. VSMCs were found to be essential for plaque stability because they are the main source of collagen for the fibrous cap of plaques. VSMCs are responsible for plaque stability, giving the tensile strength and preventing the plaques from rapture. Consequently, loss of VSMCs by any means is a negative scenario, leading to multiple severe outcomes: thinning of the fibrous cap, the formation of necrotic core, and calcification [127]. There are three main routes of VSMCs death: apoptosis, necrosis, and autophagy; each has its consequences and impact on atherosclerosis propagation [128]. Accordingly, multiple strategies have been developed to address these issues. To prevent VSMCs from apoptosis, different caspase inhibitors were tested. Rapamycin-loaded gel-like polymeric nanoparticles were synthesized from a mixture of N-isopropylacrylamide, N-vinyl pyrrolidone, and PEGylated maleic polymers [129]. Rapamycin has dual-modality, preventing VSMCs apoptosis and proliferation. Nanoparticles in vitro showed a 20% decrease in VSMCs proliferation rates compared to free rapamycin. Docetaxel-loaded lipid nanoparticles in in vivo models show a significant decrease in apoptotic VSMCs (24% versus 70% in control). LDE-DTX reduces caspase 3 by 70%, caspase 9 by 50%, and Bax by 50%. However, according to other findings, VSMCs can turn from apoptosis to necrosis when caspase 3 is inhibited [130]. From this point of view, dealing with apoptosis may seem unfavorable. Though, a recent finding proposes a new kind of necrosis inhibitor called NecroX-7 which helps to diminish the necrotic core of plaques [131]. Due to its novelty, no nanoformulated version of NecroX-7 has been reported so far. 

### 4.4. Inhibition of Pro-Inflammatory Factors 

Since atherosclerosis is a chronic inflammatory process, a lot of pro-inflammatory factors and cytokines are secreted by the endothelium and affected cells [132]. While keeping VSMCs alive is a good strategy for the treatment of aged plaques, their proliferation is a key factor in the development of the early stages of atherosclerosis. VSMCs proliferation contributes significantly to post-angioplasty restenosis [133]. VSMCs proliferation and migration are reported to be controlled by several growth factors, adhesion molecules [134], and pro-inflammatory factors [135]. Thus, inhibition of parts of these targets can potentially be beneficial for plaque growth prevention. DTX was reported to significantly reduce the expression of NF-κB, TNF-α, IL-1β, and IL-6 [114]. Moreover, DTX-loaded NPs caused a decrease in macrophage infiltration inside adventitial layers and an overall reduction in inflammation. PLGA-PEG nanoparticles, carrying the Ac2-26 peptide, reduced necrotic area, stabilized fibrous cap, and suppressed oxidative stress in hypercholesterolemic mice (Table 2) [107]. Improving these parameters can diminish overall inflammation. An interesting approach to lower inflammatory cytokines levels was presented in [106]. Gao et al. used NPs covered in a macrophage membrane that can naturally adsorb cytokines such as TNF-α and IL-1β via interaction with membrane antigens (e.g., TNFR2, CD36, and CCR2). While proof-of-concept was only shown in vitro, in vivo studies revealed promising results of such NPs. Other biomimetic liposomes were used to deliver pioglitazone, peroxisome proliferator-activated receptor γ (PPARγ) agonist, to reduce pro-inflammation factor content inside plaques (Table 1 and Table 2) [20]. While the effect on cytokine levels was significant, the overall effect on plaque stability was moderate. 

### 4.5. Protection of Macrophages

Transformation of macrophages from normal to diseased phenotype—foam cells—is one of the main factors of atherosclerosis development. Pathogenic transformation occurs due to the accumulation of oxidized low-density lipids (oxLDL) inside macrophages. Accumulation of foam cells leads to inflammation, cell death, and formation of the necrotic core. Thereby, the prevention of oxLDL uptake may be beneficial. Moghe et al. have synthesized sugar-based amphiphilic nanoparticles with average sizes in the range of 100–400 nm that can block scavenger receptors by mimicking oxLDL [57]. This strategy proved to be effective in the prevention of foam cell formation, resulting in reduced inflammation, lower lipid accumulation, and absence of artery occlusion. However, the following strategy is the most suitable for the early stages of atherosclerosis. Targeted delivery of liver X receptor (LXR) via PLGA-b-PEG nanoparticles showed high efficiency in reducing CD68-positive macrophages in the atherosclerotic area (Table 2) [46]. CD68^+^ staining showed 20% greater reduction in macrophage content in aortic plaques (to a total 50% decrease compared to control). LXR affects cholesterol efflux in macrophages, enhancing efferocytosis, and reducing inflammation. While direct administration of LXR results in hepatic steatosis, targeted delivery with NPs has proven to be viable. The researchers have managed to achieve a high loading rate (65.4 ± 5.8% from the initial drug input weight) and almost complete release (84.4% over 8 days). A similar case based on the prevention of cholesterol efflux in macrophages was achieved by silencing miR-33 micro-RNA (Table 2) [48]. miR-33 causes a lot of atherosclerosis manifestations because it is responsible for lipid metabolism, fatty acid oxidation, and autophagy of macrophages [136]. However, some additional positive changes were observed in mice treated with NPs loaded with miR-33 [48]. It is crucial to mention, that the authors managed to achieve both a high loading rate of 88% and a complete release of miR-33 (>96%). The full release was possible by the use of a pH-dependent carrier, which entirely dissolves under acidic conditions of endosomes. The authors reported reduced necrotic area, reduced amount of lesional macrophages, a lowered MMP 3 level, and increased VSMCs numbers in aortic plaques. 

Another common cause of macrophage death is via exposure to ROS. ROS also can cause apoptosis of VSMCs. Thus ROS-scavenging nanoparticles can play a dual-protective role in atherosclerosis treatment. The concept was proven with β-cyclodextrin-based nanoparticles (Table 2) [44]. This polysaccharide was conjugated with two ROS-eliminating substances: tempol, mimicking superoxide dismutase, and phenylboronic acid pinacol ester, eliminating hydrogen peroxide. On a standard ApoE^−/−^ mouse model it was shown that passive accumulation of particles is enough to lessen the main manifestation of atherosclerosis. Results such as reduced cholesterol level, lowered necrotic core volume, and thicker fibrous cap were obtained. 

### 4.6. Reduction in Macrophage Proliferation

The reduction in macrophage proliferation seems to be a promising approach. In developed plaques, macrophage proliferation contributes to 80% of local macrophage accumulation [137]. Statins are known to suppress the proliferation of different cell types, including macrophages [138]. Oral statins treatment, along with the intravenous injection of a high-density lipoprotein nanoparticle loaded with simvastatin, showed promising results (Table 2) [108]. This approach was especially effective for plaque inflammation reduction (by 43% in terms of plaque area) following a stroke or myocardial infarction, with the known recurrence rate within 3 years for these states being ~20% [139]. The natural accumulation of lipid nanoparticles in macrophages accompanies conventional statin treatment and helps to reduce the treatment time significantly. A similar approach was based on cyclodextrin NPs loaded with simvastatin (Table 1) [45]. Kim et al. showed that the affinity of cyclodextrin to cholesterol can act both as scavenging modality and release stimuli [45]. Simvastatin was replaced with cholesterol and eliminated from plaques providing a dual-treatment approach. It was also proven to be promising for both the prevention and treatment of advanced plaques.

### 4.7. Efferocytosis Mediation

In recent years several studies have shown that efferocytosis may be inhibited in atherosclerotic lesions [140,141]. CD47 is mainly involved in this process [142]. However, direct inhibition of CD47 leads to anemia through elevated red blood cells elimination by phagocytosis [66]. Thus, the targeted delivery of CD47 inhibitors could be a potential solution to this issue. In [109] efferocytosis was restored through a passive delivery of a Src homology region 2 domain-containing phosphatase-1 (SHP1) inhibitor via single-walled carbon nanotubes (Table 2). SHP1, which is known to suppress phagocytic functions, is activated by the CD47-SIRPα signaling pathway. Passive accumulation of nanotubes was established via the natural way by Ly-6Chi monocytes, which further underwent differentiation to lesional macrophages. As a result, reduced lesion area, smaller necrotic core, lower cell debris accumulation within plaques, and, more importantly, no adverse side effects on red blood cells were observed. However, the choice of single-wall carbon nanotubes (SWNT) as carriers are doubtful since a great deal of data show their potential toxicity [143,144]. Moreover, the release rate from SWNTs was not so high (~13% over 7 days). Similar results were achieved by Tao et al. [47] by delivering siRNA, inhibiting Ca^2+^/calmodulin-dependent protein kinase γ (CaMKIIγ), thus promoting efferocytosis (Table 2). Nanoparticles were of a complex structure, with a PLGA core and lipid-PEG shell, conjugated with target peptide. This allowed the particles to effectively accumulate in plaques and macrophages, avoiding side effects. The overall stabilization of the plaque was attained by simultaneous reduction in necrotic core (due to more effective efferocytosis) and thickened fibrous cap. 

### 4.8. Preventing Low Shear Stress Consequences

Despite the inflammatory nature of atherosclerosis, there is a reason for inflammation to start and it is connected with shear stress in different vessel regions. For now, it has already been proven that shear stress can regulate endothelium behavior [4,145]. Recently it was reported that shear stress in vessels can regulate the expression of microRNA (miRNA) in endothelial cells [146]. Such miRNAs are called mechano-miRNA, and now the whole family of these molecules has been found. Overexpression of mechano-miRNA leads to alteration of the endothelial cell’s life cycle, apoptosis, and inflammation. Some of the miRNA, such as miR-712 or miR-205 can target tissue inhibitors of metalloproteinase 3 (TIMP3) which leads to the activation of MMPs [147]. Hyperactivity of MMPs leads to lesion, endothelial inflammation, and hyperpermeability. One of the pioneer groups in the development of anti-miRNA therapies, led by professor Hanjoong Jo, suggested several nanocarriers to deliver anti-miR (Table 2) [57,79]. Both of these systems were targeted through VCAM1-targeting peptides. Since nanoparticles were targeted through VCAM1, it was necessary to avoid nanoparticles elimination by macrophages. In this sense, an important result was the establishment of the fact that 5 nm gold nanoparticles are the best for these purposes (Table 1) [79]. Total amounts of 10, 20, and 50 nm Au nanoparticles mostly accumulated in the liver and other mononuclear phagocyte system (MPS) organs.

### 4.9. Combined Approaches

While some studies are dedicated to the treatment of a very narrow pool of atherosclerosis symptoms, some nanoformulations deliberately or occasionally show a broad spectrum of anti-atherosclerotic activity. A rather complex formulation based on oxidation-sensitive chitosan oligosaccharide nanoparticles coated in macrophages membrane (MM) and loaded with atorvastatin was shown to be efficient for multiple tasks (Table 1) [106]. The authors managed to deliver a statin drug to plaque proximity via passive accumulation of NPs in lesion macrophages. More interestingly, MM possessing antigens on its surface, was able to sorb inflammatory cytokines from plaques. Additionally, this formulation caused proliferation of VSMCs, elevated collagen amount in fibrous cap, and, simultaneously, reduced endothelial proliferation. Although the effects considered individually were small, the net effect on plaque stability was significant.

## 5. Magnetic Nanocarriers for Atherosclerosis Therapy

The magnetic field application is a convenient tool to increase the local concentration of a therapeutic agent and to mediate stimuli-responsive therapy (Figure 2a) [148]. The magnetic field is harmless in a wide magnetic induction value range, non-invasive, and has no energy dissipation in biological cells and tissues in comparison with electromagnetic waves [149]. The transduction of magnetic field energy by magnetic particles into heat or mechanical forces has a lot of possible therapeutic applications and plays a crucial role in modern biomedicine. One of the main tasks of magnetic field-based therapy is to choose the correct magnetic carrier for the drug and suitable physical stimuli to release and activate the drug or generate the heat and mechanical stress in the particular site. The use of magnetic particles has numerous benefits such as (i) the possibility to guide it to the tissue or organ in blood flow, (ii) various synthetic strategies for the rational design of carriers for a particular therapeutic need, thus enabling customization of the system and (iii) energy conversion into the heat or mechanical stimuli when applying a magnetic field.

To date, magnetite is one of the most frequently used magnetic materials for magnetic field-based biomedicine. Its excellent biocompatibility [150], a large number of synthetic strategies to obtain magnetite of various shapes (cubic, spherical, ellipsoidal, etc.) and sizes (from 10 nm up to 10 µm) [91], and significant magnetic response (saturation magnetization of bulk magnetite is 92 emu/g) determine a routine use in a large number of papers. Moreover, the availability of magnetite-based FDA approved Endorem™ (AMAG Pharmaceuticals, Inc., Waltham, MA, USA; Guerbet S.A., Aulnay-sous-Bois, France), Feridex^®^ (Bayer HealthCare Pharmaceuticals Inc., Whippany, NJ, USA), and Resovist^®^ (Bayer Schering Pharma AG, Bergkamen, Germany) medications prove the promising clinical potential for further investigation.

Although the magnetic field-based strategy was utilized for various biomedical applications, there are only a few works dedicated to the use of magnetic particles in atherosclerosis therapy. Magnetic resonance imaging (MRI) represents the largest field of magnetic particles application in the context of atherosclerosis and inflammatory diseases [151]. Superparamagnetic iron oxide nanoparticles (SPIONs) act primarily to alter T2 values of water protons surrounding the particles. When SPIONs are localized in the tissue in the presence of an external magnetic field, the magnetic moments of the particles align to create heterogeneous field gradients that affect water protons diffusion. Dipolar coupling between the magnetic moments of water protons and the magnetic moments of particles causes efficient spin dephasing and T2 relaxation leading to a decrease in signal intensity [152].

In contrast to imaging, Liu et al. showed the efficiency of dual magnetothermal–photothermal treatment based on Fe_3_S_4_ particles for the elimination of infiltrating inflammatory macrophages [153]. In vivo experiments on ApoE^−/−^ mice suggested that the combination of photothermal therapy and magnetic hyperthermia was effective in the elimination of inflammatory macrophages and in the further inhibition of atherosclerosis and arterial stenosis formation. Although the examples of magnetothermal therapy for atherosclerosis are still at an early stage, several points should be discussed. The magnetothermal issue has a major disadvantage, namely, off-target heating of adjacent tissues which could cause problems much bigger that will not be compensated by treatment, especially near temperature-sensitive organs. From this point of view, the design of magnetic nanocarriers with low Curie temperatures, around 42–43 °C, and accurate targeting of particles to the tissue of interest by specific markers or magnetic guides can significantly increase the effectiveness of magnetothermal treatment of atherosclerosis [154].

The next strategy involves a magnetic drug delivery system that allows magnetic-guided drug targeting to inflammatory tissue. Matuszak et al. showed the feasibility of magnetic drug targeting to arterial vessels in vivo and the therapeutic efficacy of SPIONs conjugated with an anti-inflammatory drug dexamethasone phosphate (SPION-DEXA) [155]. Although efficient magnetic targeting was demonstrated, the authors claimed that DEXA seemed to be a non-optimal drug in the case of atherosclerosis. Nevertheless, to address the issue of effective magnetic drug delivery the large magnet was used followed by the dissection of tissue to ensure the close distance between the magnet and vessel.

The biocomposite based on magnetite nanoparticles serves as a promising platform for encapsulation and multimodal (imaging and treatment) action for atherosclerosis. Dong et al. aimed at simultaneous imaging and treatment of atherosclerotic plaques [156]. Their data suggested that magnetite-based composite could achieve the aim of simultaneous detection and therapy of atherosclerotic plaques, according to co-delivery of SPIONs and paclitaxel to macrophages. Wu et al. constructed a self-driven bioinspired multimodal nanosystem for early detection and treatment of atherosclerosis [157]. The Fe_3_O_4_ magnetic nanoclusters served as a core, sequentially embedding anti-inflammatory drug simvastatin and decorating it with targetable apolipoprotein A-I mimetic 4F peptide. The magnetic composite exhibits excellent anti-atherosclerotic effects by alleviating inflammation and oxidative stress as well as promoting cholesterol efflux via reverse cholesterol transport pathways. Another example of the non-target coating was demonstrated by Zhang et al. [158]. The authors demonstrated that DNA-coated PEGylated SPIONs effectively accumulate in the macrophages of atherosclerotic plaques after the intravenous injection into ApoE^−/−^ mice.

## 6. Utilization of Theranostic Agents

The theranostic approach represents a field of drug delivery that allows one not only to access imaging and visualize pathology sites but also to produce therapeutic action at the same time [159]. In turn, drug delivery systems (DDSs)—whether of inorganic (nanoparticles) or organic (polymeric particles, micelles, liposomes, etc.) nature—can enable precise control over the bloodstream circulation time, required drug dosage, accumulation selectivity. Moreover, DDSs also reduce drug toxicity and off-target effects [160], thus its application in theranostics is entirely justified. DDS targeting can be achieved by atherosclerosis-associated receptors like ανβ3 integrin, stabilin-2, CD44, etc. which was previously discussed in the nanocarriers-based drug targeting principles section. Imaging can be achieved in various ways including fluorophore-, radionuclide- or other stimuli-responsive molecule bearing DDSs [161], DDS stimuli-responsive by itself [162], or its combination for imaging resolution improvement [163]. The therapeutic action of the theranostic platform can be due to the chemical [164], physical [165], or combined [166] impact. To date, the most common theranostic approaches towards the treatment of atherosclerosis are photodynamic therapy (PDT) [167] and photothermal therapy (PTT) [168], even though there are other equally promising approaches. In the following section, we discuss recent progress in the field of theranostic platform addressing atherosclerosis development and its clinical application.

### 6.1. Photodynamic Therapy (PDT)

Photodynamic therapy or PDT uses compounds capable of being activated by light at a particular wavelength with subsequent production of cytotoxic reactive oxygen species or ROS (Figure 2b), which can then damage macrophages. In the case of atherosclerosis, targeting macrophages is often implemented due to its participation in disease pathogenesis, namely internalization of LDL leading to foam cell formation, inflammation, and atherosclerotic plaque formation.

There are two mechanisms of photosensitizer action, both of them start from the absorption of light of a particular wavelength, followed by photosensitizer excitation due to transition from ground singlet (PS singlet) to excited singlet state (PS* singlet). The first mechanism implies the presence of excited photosensitizer (PS* singlet) that undergoes “so-called” relaxation, which is the process of non-emissive transition to excited triplet state (PS* triplet). PS* triplet can transfer proton/electron to biomolecules resulting in the formation of chemically active free radicals. The second mechanism involves energy from the excited triplet state (PS* triplet) transfer to the oxygen molecule in the ground triplet state (O_2_ triplet), resulting in the formation of excited singlet state oxygen (O_2_* singlet), which is possible due to the presence of two electrons with the same spins. Photosensitizers can generate chemically active and toxic radicals when radiated by the light of a particular wavelength. Additionally, all photosensitizers are fluorescent, so they can be used as theranostic agents without additional dyes [169]. 

Conventionally used photosensitizers are usually porphyrin derivatives [170,171] since the very first compound explored to be photosensitizer was a hematoporphyrin derivative (consisting of porphyrin dimers and higher oligomers chemically cross-linked). Several clinically approved porphyrin-based therapeutics exist nowadays, e.g., Photofrin^®^, Radachlorin^®^, Levulan^®^, Visudyne^®^, Foscan^®^, Verteporfin^®^, etc. There is a large variety of other chemicals used as photosensitizers in PDT, e.g., taxaphyrins, thiopurine derivatives, phthalocyanines, etc. For example, 5-aminolevulinic acid (ALA)—which is the precursor of protoporphyrin IX—has been used for the treatment of atherosclerosis in an animal model [172]. While being extremely effective in ROS production, these small molecules have a low selectivity of accumulation and, thereby high toxicity and off-target effects, not to mention low bloodstream circulation times. 

It was demonstrated by Han et al. [173], that PDT notably enhances the cholesterol efflux from macrophages alongside the induction of autophagy through the production of ROS. Additionally, to minimize some of the PDT and drug disadvantages, photosensitizer encapsulation into DDS is employed to a much greater extent. For example, McCarthy et al. encapsulated chlorine derivative, which has a similar structure to porphyrin, into iron oxide nanoparticles [174]. This nanoformulation, possessing high biocompatibility and biodegradability, was further coated with dextran to achieve macrophage targeting and exhibited light-dependent cytotoxicity. To achieve better selectivity of the theranostic platform, cathepsin-B activatable photosensitizer was used [175]. Cathepsin-B is a protease produced by macrophages; thus, the selectivity of this platform is due to the combination of increased accumulation in macrophages, selective production of pro-photosensitizer in a particular cell type, and its activation by light. In another study, Spyropoulos-Antonakakis et al. used zinc phthalocyanine encapsulated in polymeric dendrimer PAMAM (poly(amidoamine)) [176]. PAMAM prevents drug aggregation on endothelial cells, thus, preparation of the nanoformulated drug was advantageous compared to the free drug. It was shown by Kim et al. [177] that nanocomposite comprised of chlorin e6 (Ce6)-hyaluronic acid (HA) conjugate can be used as a photosensitizer, with additional selectivity achieved via binding of HA to the macrophage membrane. Wennink et al. [178] developed a m-tetra(hydroxyphenyl)chlorin-loaded benzyl-poly(ε-caprolactone)-b-methoxy poly(ethylene glycol) micelle-based nanodrug to both improve loading capacity and provide rapid release due to higher lipase activity in RAW264.7 macrophages. However, the authors could not provide prolonged circulation of the nanoformulation in the bloodstream, and spontaneous drug release occurred, thus further optimization is needed to achieve greater selectivity in this case. An interesting approach was demonstrated by Lu et al. [179], in which utilizing surface plasmon resonance of nanoformulated selenium nanoparticles (SeNPs) induced both redshift, absorption bands broadening, and increased the intensity of photosensitizer rose bengal (RB). It was shown that RB chemically bonded to SeNPs was much more effective in the generation of singlet oxygen (^1^O_2_) than free RB. To provide macrophage targeting, SeNPs were coated with HA and folic acid conjugate connected by ethylenediamine linker (HA-EDA-FA), with the folate receptor beta (FR-β), which is overexpressed by activated macrophages in inflammatory disorders, as the target. Therefore, photosensitizer effectiveness can be drastically increased by targeting moiety.

In summary, PDT represents a theranostic strategy enabling both the visualization of atherosclerotic lesion localization and macrophage ablation. Photosensitizers can be designed from scratch or existing photosensitizers can be chemically modified to achieve greater therapeutical performance. Improvements can include:deeper light penetration into the tissues;greater selectivity due to chemical moieties allowing the drug to be activated only in macrophages;utilization of drug delivery systems as additional targeting modality;higher sensitivity and ^1^O_2_ production due to improved spectral characteristics;lower toxicity;the reduced amount of off-target effects of the drug due to drug encapsulation.

### 6.2. Photothermal Therapy (PTT)

Photothermal therapy (PTT) utilizes the “so-called” photothermal transduction agents (PTAs), which can transform the energy of light into heat, resulting in a local temperature increase (Figure 2c). Ideal PTAs provide maximum local heating, minimizing off-target effects and damage to healthy tissues. To provide the local heating, the rational design of PTAs with the main absorption peak in the tissue-transparent window between 750 and 1350 nm is implemented (750–1000 nm is NIR-I and 1000–1350 nm is NIR-II window). Moreover, ideal PTA has to have high photothermal conversion efficacy (PCE), meaning that its main absorption peak does not overlap with the tumor background and surrounding healthy tissues. PTA can be of organic (cyanines, porphyrins, phthalocyanines, BODIPY, croconaine, semiconducting polymers, etc.) or inorganic (noble metals, graphene, black phosphorus, boron nitride, MXenes, chalcogenides, metal oxides, etc.) nature. Inorganic PTAs usually have a higher PCE and better photothermal stability. In turn, organic PTAs are often much more biocompatible and biodegradable. PDT usage allows precise tissue targeting since laser irradiation of a particular wavelength and dosage is needed to start the process. Even though PTT is very similar to PDT in some aspects, its damaging mechanisms are different.

Cell death triggered by PTT can be of necrotic (most of the time) or apoptotic nature. Necrosis manifests itself in the loss of plasma membrane integrity, which is followed by the release of intracellular material, including damage-associated molecular patterns (DAMPs), in the extracellular matrix. These processes, in turn, trigger inflammatory and immunogenic responses, which is an undesirable outcome for a theranostic drug. In contrast, apoptosis does not result in cell membrane disruption. In the process of apoptosis, the cell generates signals for phagocytes, which internalize forming apoptotic bodies in response. Thus, apoptosis induction is much more profitable in this case.

Considering PTT as a treatment of atherosclerosis, the simplest system consisting fully of inorganic materials was presented in the work of Lu et al. [180], who developed Cu_3_BiS_3_ nanocrystals showing intense absorption in the NIR region and a high conversion efficiency of 59%, demonstrating an amazing photothermal effect. Moreover, it was shown that these nanoparticles can efficiently kill macrophages in both in vitro and in vivo experiments. Cu_3_BiS_3_ nanocrystals can also be used as an efficient contrasting agent (CT) for atherosclerotic plaque imaging. However, as in the case of PDT, targeting it is needed to provide local treatment. For example, Gao et al. [181] showed that CuS nanoparticles coupled to an antibody TRPV1 channel selectively bound to activated macrophages and forced TRPV1 channels to open upon NIR light exposure. Further Ca^2+^ efflux activates autophagy and inhibits foam cells formation. The nanocarrier can be both an imaging and therapeutic agent, which was demonstrated and thoroughly studied by Qin et al. [165]. In particular, gold nanorods were synthesized and used as photothermal agents. Significant macrophage-killing efficacy at low doses under irradiation by IR light was demonstrated in vitro on Ana-1 cells and in vivo on ApoE^−/−^ mice, although no targeting moiety was used in this study. 

In summary, PTT represents a theranostic strategy enabling both visualization of the atherosclerotic lesion and thermal ablation of macrophages as a treatment of atherosclerosis. Photothermal activity is usually observed for optically active nanoparticles as, for example, noble metal-based nanostructured materials. These structures could be chemically modified to achieve greater therapeutical performance, e.g., deeper light penetration into the tissues and greater selectivity. Utilization of antibodies, aptamers, antagonists, and small molecules as an additional targeting modality, can help to achieve lower toxicity and off-target effects due to selective accumulation in the site of the atherosclerotic lesion.

### 6.3. Other Theranostic Approaches

Although photodynamic and photothermal therapy represents the two most commonly used theranostic approaches, there are several alternatives. In their work, Harel-Adar et al. [182] developed an approach to modulate cardiac macrophages after myocardial infarction (MI) to a reparative state by using phosphatidylserine (PS)-presenting liposomes with MRI modality as mimicry of anti-inflammatory effects of apoptotic cells. Intravenous injection of PS-presenting liposomes affected the extent of angiogenesis as well as infarct size and left ventricle (LV) remodeling after MI. In another work, superparamagnetic nanoparticles were conjugated with a Gd-based imaging moiety and peptidomimetic vitronectin antagonist to target the less abundant molecular epitope, ανβ3 integrin. The approach was further used to visualize angiogenesis in the early stages of atherosclerosis via MRI [183]. Marrache et al. synthesized complex nanostructures, constructed from high-density lipids (HDL)-based nanoparticles, whose core is composed of PLGA and quantum dots (QDs) and the shell is decorated with triphenylphosphonium (TPP) cations. They were used to target macrophages, induce apoptosis, and also to bind excess of plasma cholesterol [184]. Detection of vulnerable plaques is achieved by targeting the disruption of mitochondrial membrane potential that occurs during apoptosis due to the presence of TPP. QDs were included in the nanoparticles design to provide optical imaging. Moreover, HDL nanoparticles closely mimic endogenous HDL in cholesterol-binding properties, thus in vitro studies of these nanoparticles have shown its potential in inverse cholesterol transport. It was also demonstrated that mitochondria-targeted nanoparticles can sense apoptosis with high resolution. Nevertheless, theranostic approaches are usually restricted to the use of other imaging techniques as, for example, MRI and positron emission tomography (PET).

## 7. Conclusions and Future Prospects

Considering advanced applications of NPs in the treatment of atherosclerosis, the main opportunity lies within the new molecular agents. As mentioned, several promising molecular agents, like NecroX-7, PARP-1 [185], or cyclin-dependent kinase inhibitor 2B [186] have not yet been tested in their encapsulated forms. Preventing shear stress is now one of the most promising approaches that has not yet been utilized to its full capacity. However, it should always be kept in mind, that atherosclerosis is a long-developing disease. Preventing atherosclerosis is always more beneficial than trying to reduce the already formed plaque. Calcification of aged plaques represents a very serious issue since these plaques are almost impossible to treat with chemical drugs [187]. Recent investigation has shown that retinoic acid can affect the calcification of vessel walls [188]. In 2019 a much simpler molecule—chelator ethylenediaminetetraacetic acid (EDTA)—was loaded in albumin nanoparticles and showed promising results in decreasing calcium content in the vessel wall [189]. Since most of the inorganic components of plaques are calcium phosphates and carbonates, other similar approaches can be utilized. 

The magnetic carrier-based platform could be promising for atherosclerosis treatment. Even though there are several magnetic platforms for atherosclerosis, the further possibility of transfer from in vitro and in vivo study to clinical trial evaluation needs to be discussed. One of the main questions is whether we can use the inhomogeneous magnetic field for magnetic composite guidance? Notwithstanding, several examples of magnetic drug delivery systems exist, the two key obstacles should be solved. 

First of all, the generation of large inhomogeneous magnetic fields is challenging: the exponential decrease in magnetic field magnitude with distance plays a crucial role to achieve enough force factor in the blood vessel in case of intravenous administration. This obstacle is also connected with a large velocity and blood pressure in arterial vessels, especially in the case of atherosclerotic-damaged tissues. The overcoming of the above-mentioned issue could be found in the following processes. (I) Design of a magnetic system that can load a sufficient gradient force on a magnetic particle in a blood vessel; (II) combination of magnetic drug delivery with passive or active delivery through targeting to receptor-of-interest, and (III) design of magnetic particle with larger magnetic susceptibility to provide high efficiency of the delivery of a therapeutic drug to the atherosclerotic area.

The second one is choosing the right physical stimuli for the enhancement of drug-mediated atherosclerosis treatment. Despite the sporadic evidence of using magnetic stimuli to fight with atherosclerosis, another opportunity could be used. Magnetothermal treatment is a feature of magnetic carriers that also could provide a sufficient impact on atherosclerotic plaques. However, the rational design of a carrier must overcome the well-known problem of non-selective thermal influence on healthy tissue. The solution to this problem could be found in the efficient targeting of particles and choosing the material with low Curie temperature. The prospect of synergetic treatment of atherosclerosis could also be found from a mechanical point of view. The magnetic particles in the presence of low-frequency alternating magnetic fields provide mechanical influence on tissues. This could be of interest concerning the permeation of the drug into the plaque.

Concerning PDT, the main struggles are related to the fact that the ability of light to penetrate tissues is highly wavelength-dependent (Figure 2). For photosensitizers typically used penetration depth is less than several millimeters. One of the ways to address this issue is the use of “so-called” upconversion nanoparticles, which can emit light of a shorter wavelength in response to excitation by a longer one [190]. Thus, conventional and effective photodynamic therapeutics can be used in conjunction with upconversion nanoparticles. For example, Wang et al. [191] synthesized Ce6-loaded PEG-coated NaYF_4_ upconversion nanoparticles with absorption on 980 nm and emission on c.a. 400 nm, which is the main absorption peak of Ce6. Due to the presence of a hydrophobic linker between PEG and NaYF_4_ some hydrophobic drugs can be physically adsorbed on the surface of these particles. Moreover, the developed theranostic platform was evaluated in in vitro and in vivo experiments, which makes its potential translation to the clinics more feasible. Another approach towards the solution of the tissue penetration problem is the chemical modification of conventional photosensitizers [192] and the rational design of novel [193] photosensitizers. For example, Lu et al. [194] summarize some modification strategies for the rational design of red/NIR boron-dipyrromethene (4,4-difluoro-4-bora-3a,4a-diaza-s-indacene, BDP or BODIPY) fluorescent dyes, which can later be used as an imaging modality in PDT. Thus, achieving higher tissue penetration depths represents an important task in PDT which can be addressed by several approaches, namely, chemical modification of existing photosensitizers for the achievement of redshift, rational design of novel photosensitizers having high absorption intensities, and the utilization of upconversion nanoparticles capable of being activated by light in red/NIR region and then emitting light of a shorter wavelength.

## Figures and Tables

**Figure 1 pharmaceutics-12-01056-f001:**
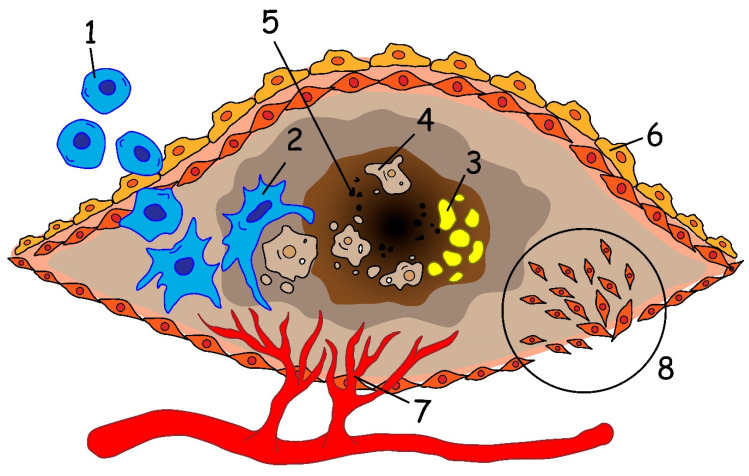
The main routes of atherosclerosis development and plaque growth. (**1**) Monocytes are being recruited to a developing plaque, transforming (**2**) to macrophages. Macrophages accumulate in plaque, absorb (**3**) lipids and become (**4**) foam cells. Apoptosis and necrosis of foam cells, as well as deficient efferocytosis, lead to the formation of (**5**) necrotic core inside the vessel wall. (**6**) Endothelium cells produce different pro-inflammatory factors that promote the accumulation of macrophages. Extensive plaque (**7**) neovascularization leads to intraplaque hemorrhages. Migration of (**8**) VSMCs promotes neointima growth and restenosis after grafts installation.

**Figure 2 pharmaceutics-12-01056-f002:**
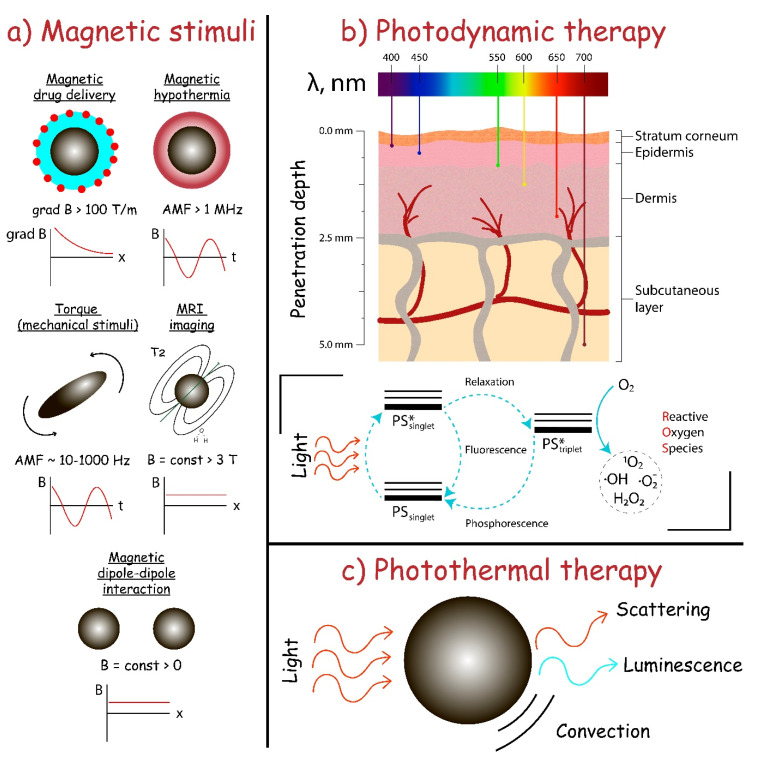
(**a**) The use of magnetic nanoparticles for therapy and imaging; (**b**) The main principle of the photodynamic therapy (PDT) approach and wavelength-dependent tissue penetration depth; (**c**) The main principle of the photothermal therapy (PTT) approach.

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
