# Peer review of "Nanoparticle-Based Approaches towards the Treatment of Atherosclerosis"

_pharmaceutics, 2020, doi:10.3390/pharmaceutics12111056_

Round 1

Reviewer 1 Report

The authors have produced an interesting review manuscript on the use of nanoparticles for the treatment of Atherosclerosis.  Although the paper is extensive there are some gaps that need to be fixed.

  1. The title in on nanoparticles but very little is discussed in the manuscript about the different nanoparticles. A section on the various nanoparticles being used in therapies not necessarily atherosclerosis, but could have a potential for its treatment, should be added.
  2. The extent of the review should be detailed. Maybe a timeframe and specifics that were considered for the review need to be included. 
  3. In the title, I feel "nanoparticles-based" should be "nanoparticle-based".

Author Response

Reviewer 1 evaluation

  1. The title in on nanoparticles but very little is discussed in the manuscript about the different nanoparticles. A section on the various nanoparticles being used in therapies not necessarily atherosclerosis, but could have a potential for its treatment, should be added.

Thank you for this remark. Following the recommendation of reviewers, we added Section 3 (Methods for the synthesis of nanoparticles) on Page 4-8, describing main approaches for the synthesis of nanoparticles for biomedical application.

  1. The extent of the review should be detailed. Maybe a timeframe and specifics that were considered for the review need to be included.

We added the timeframe statement to the abstract in lines 18-19.

  1. In the title, I feel "nanoparticles-based" should be "nanoparticle-based".

We corrected the title following your recommendation.

Reviewer 2 Report

The Authors proposed a review entitled “Nanoparticles-based approaches towards the treatment of atherosclerosis”. This paper denotes a huge study of the literature related to the proposed field of application; however, in some parts it seems a bit confused due to the very large number of information reported.

I also suggest a revision of English syntax and grammar in the manuscript. There are some typing mistakes or missing verbs in some cases.

Here are the issues that I propose to authors:

  1. An abbreviation list could be added to the manuscript where the guidelines indicate that it is possible. Generally, it is among the Abstract and the Introduction section. An abbreviation list could contain, for example, VSMC, LDL, oxLDL, HA, NP, PDGF, MM, SPION, MRI, PDT, PTT, PTA and more others used in the manuscript.
  2. There are some concepts that are well-known, but only for readers that are expert in the related fields. However, readers belonging from different areas deserve additional references. For this reason, a reference could be added after the first sentence, at line 22.
  3. Line 30. “… while in regions with laminar flow, the occurrence of plaques is very low”. My suggestion is the addition of “the” in this sentence and the substitution of “are” with “is” since “occurrence” is singular.
  4. Missing reference after the sentence at line 35.
  5. Line 72. The verb “is” is missing in the sentence “…treatment strategy for now … still a surgical removal”.
  6. Line 76. “their side effects-to-effectiveness balance of far from…”. Maybe “of” should be substituted with “is”.
  7. Line 81. “most nanoparticles are “eaten” by macrophages but macrophages accumulate in plaques”. Please develop this last concept with another sentence.
  8. Table 1. Could you specify if “passive accumulation” means that there is a cellular uptake of these molecules?
  9. In Table 1 and Table 2, information is given about the nanoparticles that have effectiveness in vitro and in vivo, respectively. However, there is some missing information that I think could be easily found due to the presence of the references in the tables. The average mean size of nanoparticles could be added, as well as the encapsulation efficiency (average) of the molecules that are included in the NPs.
  10. Another information could be added in the manuscript of in these tables: the methods of production of these particles. It is very important to report information about the methods; indeed, there are several and the way the NPs are produced implies different achieved properties of the final products.
  11. The absence of information about NP mean size and drug EE is also evident in the manuscript, for example at Line 190. Same observation in Line 212. The definition as NPs does not imply that they have mean dimensions compatible with cell uptake. It could be better to specify this data.
  12. There should be definition of nanoparticles, specifying whether they are lipidic nanoparticles or liposomes, for examples. Liposomes are particular lipidic systems made of a double layer; not always, lipidic nanoparticles are liposomes. It depends on their structure, for example the presence of a double layer.
  13. Line 244. “accumulation of foam cells leading to inflammation, cell death, and formation of necrotic core”. Here the verb is missing.
  14. Line 250. “showed high efficiency”. Could you quantify this efficiency in some manner?
  15. Line 258. A reference could be added here.
  16. Line 269. Replace “lessening”.
  17. Line 271. “alter the proliferation”. In which manner? Is there a reduction?
  18. Line 273. “showed promising results”. What, in particular?
  19. Line 278. “Kim et al.” there is no reference after this. Or, does it refer to ref 43?
  20. Line 337. Again, there is no information about size even if the shape and size are mentioned.
  21. Line 342. “there is only a few works”. Did you cite them?
  22. Line 353. Cite table 2 after talking of ApoE.
  23. Line 356. “infancy”. I think this should be replaced with something like “at the beginning” or “is still at early stage”.
  24. Line 358. Postpone noun as “much bigger trouble”.
  25. Line 408. I would say “at a particular wavelength”.
  26. Line 435. It seems that ROS has not been defined in the manuscript. Add it also to the abbreviation list.
  27. Line 472. “utilized so-called”. Add “the”: utilizes the so-called…”
  28. I would call paragraph 6 as “Conclusions and future prospects”.
  29. Line 544. “should be born in mind”. Maybe “burnt”? However, I would use another expression.
  30. Should the Reference section be numbered? Check it in the manuscript guidelines.

Author Response

Reviewer 2 evaluation

  1. An abbreviation list could be added to the manuscript where the guidelines indicate that it is possible. Generally, it is among the Abstract and the Introduction section. An abbreviation list could contain, for example, VSMC, LDL, oxLDL, HA, NP, PDGF, MM, SPION, MRI, PDT, PTT, PTA and more others used in the manuscript.

We added a list with the most frequent abbreviations on Pages 12-13.

  1. There are some concepts that are well-known, but only for readers that are expert in the related fields. However, readers belonging from different areas deserve additional references. For this reason, a reference could be added after the first sentence, at line 22.
  2. Missing reference after the sentence at line 35.
  3. Line 258. A reference could be added here.
  4. Line 278. “Kim et al.” there is no reference after this. Or, does it refer to ref 43?
  5. Line 342. “there is only a few works”. Did you cite them?

We apologize for the inconvenience, appropriate references were added.

  1. Line 30. “… while in regions with laminar flow, the occurrence of plaques is very low”. My suggestion is the addition of “the” in this sentence and the substitution of “are” with “is” since “occurrence” is singular.
  2. Line 72. The verb “is” is missing in the sentence “…treatment strategy for now … still a surgical removal”.
  3. Line 76. “their side effects-to-effectiveness balance of far from…”. Maybe “of” should be substituted with “is”.
  4. Line 244. “accumulation of foam cells leading to inflammation, cell death, and formation of necrotic core”. Here the verb is missing.
  5. Line 269. Replace “lessening”.
  6. Line 356. “infancy”. I think this should be replaced with something like “at the beginning” or “is still at early stage”.
  7. Line 358. Postpone noun as “much bigger trouble”.
  8. Line 408. I would say “at a particular wavelength”.
  9. Line 435. It seems that ROS has not been defined in the manuscript. Add it also to the abbreviation list.
  10. Line 472. “utilized so-called”. Add “the”: utilizes the so-called…”
  11. I would call paragraph 6 as “Conclusions and future prospects”.
  12. Line 544. “should be born in mind”. Maybe “burnt”? However, I would use another expression.
  13. Should the Reference section be numbered? Check it in the manuscript guidelines.
  14. Line 353. Cite table 2 after talking of ApoE.
  15. Line 337. Again, there is no information about size even if the shape and size are mentioned.

We apologize for these mistakes and grateful to the reviewer for pointing out to them. All mistakes were fixed and the manuscript was subjected to another round of proofreading. 

  1. Line 81. “most nanoparticles are “eaten” by macrophages but macrophages accumulate in plaques”. Please develop this last concept with another sentence.

We rewrote this sentence to better explain our idea (Lines 79-84). 

  1. Table 1. Could you specify if “passive accumulation” means that there is a cellular uptake of these molecules?

This statement was indeed unclear and we replaced it with “Phagocytosis by macrophages” in both Tables.

  1. In Table 1 and Table 2, information is given about the nanoparticles that have effectiveness in vitro and in vivo, respectively. However, there is some missing information that I think could be easily found due to the presence of the references in the tables. The average mean size of nanoparticles could be added, as well as the encapsulation efficiency (average) of the molecules that are included in the NPs.
  2. The absence of information about NP mean size and drug EE is also evident in the manuscript, for example at Line 190. Same observation in Line 212. The definition as NPs does not imply that they have mean dimensions compatible with cell uptake. It could be better to specify this data.

We are grateful to the reviewer for these suggestions. We added to both Tables a column representing the average size of nanoparticles. Finding the data about loading and release rates was tricky and information was absent in some papers. Nevertheless, we added available information directly to the manuscript text; for instance, lines 336, 404, 410.

  1. Another information could be added in the manuscript of in these tables: the methods of production of these particles. It is very important to report information about the methods; indeed, there are several and the way the NPs are produced implies different achieved properties of the final products.

Following the recommendation of reviewers, we added Section 3 (Methods for the synthesis of nanoparticles) on Page 4-8, describing main approaches for the synthesis of nanoparticles for biomedical application.

  1. There should be definition of nanoparticles, specifying whether they are lipidic nanoparticles or liposomes, for examples. Liposomes are particular lipidic systems made of a double layer; not always, lipidic nanoparticles are liposomes. It depends on their structure, for example the presence of a double layer.

We checked through the manuscript and clarified these terms, for example, line 359.

  1. Line 250. “showed high efficiency”. Could you quantify this efficiency in some manner?

The sentence was added to provide additional information, Line 400.

  1. Line 271. “alter the proliferation”. In which manner? Is there a reduction?
  2. Line 273. “showed promising results”. What, in particular?

We clarified these statements in Lines 425 and 428.

Round 2

Reviewer 1 Report

The authors have made extensive revision to the manuscript and responded satisfactorily to my comments.

Reviewer 2 Report

Authors performed several variation on their paper, improving it under different technical aspects.

They answered to my issues point by point and gave proper integrations and explanations where requested. 

I also appreciate that about 60 more references have been added to the paper bibliography, improving the state of the art.

In my opinion, now this paper deserves to be published in the present form.